# The Snake Study: Survey of National Attitudes and Knowledge in Envenomation

**DOI:** 10.3390/toxins13070482

**Published:** 2021-07-12

**Authors:** George Braitberg, Vasilios Nimorakiotakis, Celene Y.L. Yap, Violet Mukaro, Ronelle Welton, Anna Parker, Jonathan Knott, David Story

**Affiliations:** 1Emergency Department, The Royal Melbourne Hospital, Melbourne, VIC 3050, Australia; jknott@unimelb.edu.au; 2Department of Critical Care, Faculty of Medicine, Dentistry and Health Sciences, The University of Melbourne, Melbourne, VIC 3010, Australia; violet.mukaro@unimelb.edu.au (V.M.); anna.parker@unimelb.edu.au (A.P.); dastory@unimelb.edu.au (D.S.); 3Emergency Department, Epworth Hospital, Richmond, VIC 3121, Australia; bill.nimo@epworth.org.au; 4Australian Venom Research Unit, The University of Melbourne, Melbourne, VIC 3010, Australia; 5Department of Nursing, Faculty of Medicine, Dentistry and Health Sciences, The University of Melbourne, Melbourne, VIC 3010, Australia; yen.yap@unimelb.edu.au; 6Methods and Implementation Support for Clinical and Health Research Hub, The University of Melbourne, Melbourne, VIC 3010, Australia; ronelle.welton@unimelb.edu.au

**Keywords:** antivenom, snakebite, management, information

## Abstract

Despite recent reviews of best practice for the treatment of Australian venomous bites and stings, there is controversy about some aspects of care, particularly the use of antivenom. Our aim was to understand current attitudes and practice in the management of suspected snake envenoming. A single-stage, cross-sectional survey of Australian emergency care physicians who had treated snake envenomation in the previous 36 months was conducted. Hospital pharmacists were also invited to complete a survey about antivenom availability, usage, and wastage in Australian hospitals. The survey was available between 5 March and 16 June 2019. A total of 121 snake envenoming cases were reported, and more than a third (44.6%) of patients were not treated with antivenom. For those treated with antivenom (*n* = 67), 29 patients (43%) received more than one ampoule. Nearly a quarter of respondents (21%) identified that antivenom availability was, or could be, a barrier to manage snake envenoming, while cost was identified as the least important factor. Adverse reactions following antivenom use were described in 11.9% of cases (*n* = 8). The majority of patients with suspected envenoming did not receive antivenom. We noted variation in dosage, sources of information, beliefs, and approaches to the care of the envenomed patient.

## 1. Introduction

Snakebite envenomation in Australia is not common, with 3000 cases, 500 hospital admissions, and an average of 2 deaths in Australia each year [1,2]. Since 2007, there have been 1054 calls to the Victorian Poison Information Centre (VPIC) relating to snake bite exposures, an average of 81 patients per year. In the 2020 calendar year, there were 128 calls regarding snake bite exposure, of which 63 patients were reported to be asymptomatic at initial call and 31 were reported to be symptomatic (all but four reporting minor symptoms), while symptoms were not reported or documented as unknown in 34 patients. Of this cohort, 72 were male, 54 female, and 2 unknown (Personal communication VPIC Medical Director). The VPIC serves a population of 6.681 million (as of September 2020); hence, the call rate is 0.0001% (Personal communication VPIC Medical Director).

A recent review of national data showed that brown snakes cause the majority of deaths in Australia, and the overall causes of death are due to cardiotoxic and coagulopathic effects [1]. Data from animal experiments [3,4,5,6], case series [7,8,9,10,11,12,13], analyses of fatal snakebites [1,14], and prospective observational studies [15,16,17] have provided information about the epidemiology and clinical aspects of snakebite in Australia in order to inform clinical practice guidelines for patients with actual or suspected envenoming across all Australian jurisdictions. However, there are no data related to whether current clinical practice adheres to available clinical guidelines.

Australia has had a long and esteemed history of snakebite research, led over a 30-year period from 1960 by Stuhan Sutherland, who also pioneered the use of the pressure immobilisation bandage for the treatment of Australian elapids and the funnel web spider (Atrax Robustus) in the late 1970s. His contribution was acknowledged in a publication in Toxicon in 2006 [18].

Our study was conducted following concerns raised during a coronial enquiry into two snakebite deaths in Victoria. In seeking expert opinion, the coroner noted a lack of consensus in treatment guidelines amongst three experts, all of whom are well respected and two of whom are well published in the area [19]. We surmised that if experts could not agree, how were front line clinicians managing snake bite and what was guiding their care? 

Some of the challenges in managing snake envenoming in Australia have included the controversy over the number of ampoules of antivenom required for treating envenoming cases [13,20,21], the role of laboratory investigation in determining the use of antivenom, and the source of information commonly used by clinicians. 

Our study was designed to gain an understanding of the “real world” experience of clinicians who have treated snake envenoming in Australia. The number of clinicians in this cohort is small, given the low volume of symptomatic calls received by our poison information services. Over a 10-year period in 1548 patients recruited from 171 hospitals in all Australian states and territories, 755 patients received antivenom, including 49 non-envenomed patients. This is less than eight patients a year [22]. Our aim was to understand current attitudes and practice in the management of snake envenoming; identify sources of information; document the type and quantity of antivenom used; identify any barriers to management; and document compliance with clinical practice guidelines to inform, strengthen, and standardise recommendations in the future. 

Our aim was to determine if there was a need for further work on information sources and we have shown that this is the case. More work needs to be done to align practice across the country. We have not set out to provide the medical expertise (and consensus) but to establish the need to do this. The study was not designed to provide clearer instructions on behaviour in emergency situations—it was to ascertain current treatment and sources of knowledge and referral

## 2. Results

### 2.1. Knowledge, Attitudes, and Practices of Snake Antivenom Use

Our findings are presented below according to the best practices noted by the Checklist for Reporting Results of Internet E-Surveys (CHERRIES) [23]. There was a total of 217 responses recorded; 107 were excluded as they were incomplete, and 110 were included for analysis. 

More than 90% of respondents were fellows or trainees of the Australasian College for Emergency Medicine (ACEM). Three percent identified themselves as rural general practitioners. Most respondents worked in major public acute hospitals with a 24-h emergency department (ED) (Table 1). All but three respondents stated that antivenom was available at their hospitals. Most respondents had been in practice for 11–20 years and were from Victoria.

Of the 110 respondents, a total of 121 snake envenoming cases were reported (Table 2). Respondents reported 14.9% of cases using professional snake identification. The three most common symptoms reported were pain, anxiety, and headache (Table 2). Respondents identified clinical presentation as the most important factor for determining the use of antivenom and cost of antivenom the least important (Table 3). Laboratory tests were conducted in 76.3% of cases; INR and APPT were the most ordered tests (Table 4).

When asked to identify barriers to management of snakebites, respondents cited training and pathology services as the two biggest barriers (Table 5). While most were confident in their ability to treat a snakebite, nearly a quarter of respondents felt that they were uncertain or did not agree that they would make the correct choice of antivenom. While the most useful sources of information were identified as state-based guidelines, most respondents would call the Poison Information Centre or a local toxicologist for advice at the time of management (Table 4 and Table 6). Current published literature was cited as the least useful in guiding management of snake envenoming cases (Table 6). 

For the 67 patients who received antivenom, more than one type of antivenom was used in a quarter of cases (*n* = 17) (tiger snake antivenom + brown snake antivenom (*n* = 14); polyvalent + brown snake antivenom (*n* = 1); polyvalent + tiger snake antivenom (*n* = 1); taipan snake antivenom + brown snake antivenom (*n* = 1)). Overall, brown snake and tiger snake antivenom (46.3%) were used more commonly than other antivenoms (polyvalent (19.3%); taipan (7.5%); death adder (4.5%); black snake (1.5%)). In this cohort, 15 (22.4%) patients received more than one ampoule (polyvalent (*n* = 2); taipan (*n* = 1); tiger snake (*n* = 6); brown snake (*n* = 3); death adder (*n* = 2); black snake (*n* = 1)). An adverse event to antivenom was reported by 8 (11.9%) respondents: three allergic, four anaphylactic, and one case of continuing pain and anxiety. 

### 2.2. Use of Stock Holdings, Usage and Wastage of Snake Antivenom

A total of 31 hospital pharmacists completed the survey. Most respondents worked in major public hospitals in New South Wales and Victoria (Table 7). About 72% (*n* = 23) pharmacist respondents stated that near-expired antivenom would be discarded after the expiry date, while other respondents would send the near-expired antivenom to local veterinary services (*n* = 5) or use it for training or research purposes (*n* = 4). 

## 3. Discussion

Over the last two decades, there has been significant controversy over the number of ampoules of antivenom required for the initial dose in treating snake envenoming in Australia. The Australian Snakebite Project (ASP) study, an in vitro venom/antivenom neutralisation study, concluded that one ampoule of tiger snake antivenom appeared to be sufficient to bind all circulating tiger snake venom [24]. On the basis of this work, product information for both tiger snake and brown snake antivenom was changed to recommend one ampoule of antivenom for the treatment for envenomation [25,26]. However, while this recommendation is based upon the average yield of venom, actual venom yields can be higher than anticipated [2], and concern has been raised that the ASP methodology did not account for outliers [27]. 

A recent coronial investigation into two fatal deaths in Victoria highlighted different opinions amongst experts about the optimal dose and type of antivenom and directed health department authorities to review and develop a consistent set of guidelines for suspected and established snake bite [19,28].

In this study, most respondents were fellows or trainees of the ACEM. This is not surprising, given most antivenom is administered in an ED setting. Clinicians reported that they felt confident in treating snakebite but not in choice of antivenom. There was very little difference in the frequency of use of the tiger snake antivenom and brown snake antivenom, and this result was similar to a previous report of 133 patients [29], with the exception of a less frequent use of polyvalent antivenom—31% versus 19.3%. 

Accurate diagnosis relies on the combination of a good history (e.g., a snake was seen, or the bite was felt), targeted examination for symptoms of envenoming (e.g., ptosis, dysarthria), and appropriate laboratory investigations (e.g., coagulation studies). In our study, clinical presentation was relied upon the most, and while laboratory investigations were identified as vital in determining the presence and/or severity of snake envenoming, access to pathology services was reported to be a major barrier to management.

While antivenom therapy can be associated with adverse reactions, our study did not identify adverse reactions as a barrier to prescribing, and our respondents demonstrated a high level of confidence (91.8%) in their ability to treat a reaction if it occurred.

A total of 39.6% of medical respondents stated that they were uncertain or disagreed that publications in peer-reviewed journals were useful in guiding the management of snake bite envenoming, possibly reflecting the conflicting recommendations. 

Challenges in managing envenoming are not limited to antivenom dosing. There are concerns about the costs of stocking high quantity of antivenoms in the hospitals. Antivenom costs AUD 347 to AUD 2320 per ampoule and has a shelf-life of 1 to 3 years. Anecdotally, the high cost of antivenoms and the low incidence of envenoming cases have caused some hospitals to stop stocking antivenom. In our study, while over a third of physician respondents (35.5.%) identified that availability was or could be a barrier to managing snake envenoming, cost was identified as the least important factor (92.7%) Similarly, pharmacists also considered cost and shelf life as the least important factors to influence their decision to stock antivenom.

### Limitations to Our Study

The results of this study are limited by the small sample size. We were unable to determine a response rate as the number of practitioners who have treated a snakebite is unknown. We piloted the survey to improve its usability and widely distributed it to our target group; however, despite this, most responses were from clinicians located in Victoria. The study was a questionnaire and respondents may have provided answers that were “expected” rather than actual, and responses were subject to recall bias. The high mortality rate (5%) may have been due to incomplete records or selection bias as clinicians who cared for a patient who died may have been more likely to contribute to the survey.

## 4. Conclusions

Our data confirm variation in management. Over one-third of respondents stated that they were uncertain or disagreed that publications in peer-reviewed journals were useful in guiding the management of snake bite envenoming. While two-thirds of respondents felt that the availability of guidelines was not a barrier, a third were uncertain or agreed they were a barrier to snake bite management. A different number and combination of antivenom was administered in the 67 patients who received antivenom, which highlights the concerns and the premise upon which the survey was conducted.

This study provides an insight into the management of snake bite as reported by those clinicians who have treated a snake bite patient over the last 36 months. The study highlights that for one in four patients, the number of ampoules administered differs from the current manufacturer guidelines. Nearly a quarter of our respondents reported that they were uncertain or did not agree that they would make the correct choice of antivenom. Multiple sources of information were accessed with least confidence provided by peer reviewed literature. Cost was not a factor in the decision to prescribe antivenom. Access to pathology was identified as a major barrier. 

## 5. Materials and Methods

### 5.1. Survey Design

A single-stage, cross sectional survey of current knowledge, attitudes, and practices of snake antivenom use amongst clinicians who have treated snakebite patients was developed. The survey gathered information of patient presentation, outcome, and treatment used for snake envenoming and access to knowledge on envenoming, including barriers to learning needs on envenoming management, attitudes, and acceptability of published guidelines. Survey questions included demographic information for subjects: gender, years of practice in medicine; institutional characteristics: region of practice and, for emergency physicians and trainees, the level of ED in which they worked according to the ACEM classification [30]. Hospital pharmacists were also invited to complete the survey about the antivenom availability, usage, and wastage in Australian hospitals.

### 5.2. Participants and Recruitment

The survey was piloted with several representatives of the target audience, seeking feedback on appropriateness of the questions and usability. Study data were collected and managed using REDCap (Research Electronic Data Capture) hosted at the University of Melbourne. REDCap is a secure, web-based software platform designed to support data capture for research studies [31,32]. Survey invitations were distributed with a Plain Language Statement or email introduction to key networks such as members of the ACEM; The Society of Hospital Pharmacists of Australia (SHPA); clinicians who provide emergency care at University of Melbourne affiliated hospitals; and professional contacts of the study investigators, The Australasian College for Emergency Medicine newsletter, website, and twitter. Survey completion was regarded as implied consent. The survey was completely anonymous, with no identifiers recorded from respondents. Approval was obtained from the University of Melbourne Medical Education Human Ethics Advisory Group as a Minimal Risk Project (Ethics ID: 1853412.1).

### 5.3. Data Analysis 

Descriptive analyses were used to characterise study subject characteristics.

## Figures and Tables

**Table 1 toxins-13-00482-t001:** Participant demographics (*n* = 110).

Demographic Characteristics	No.	(%)
Accreditation		
ACEM ^1^ fellow	91	82.7
ACRRM ^2^ fellow	2	1.8
ACEM registrar	6	5.5
ACEM Trainee	8	7.3
Registrar (not declared)	1	0.91
Rural general practitioner	1	0.91
ACCRM registrar	1	0.91
Years of practice in medicine		
0–10	28	25.5
11–20	44	40.0
>20	38	34.5
Main state of practice		
Australian Capital Territory	1	0.91
New South Wales	14	12.7
Northern Territory	2	1.8
Queensland	12	10.9
South Australia	8	7.3
Tasmania	3	2.7
Victoria	60	54.5
Western Australia	9	8.2
Other	1	0.9
Emergency department ACEM classification		
Level 1: within a designated area of a remote or rural hospital	7	6.4
Level 2: part of a secondary hospital	8	7.3
Level 3: part of a major regional, metropolitan, or urban hospital	47	42.7
Level 4: part of a large, multifunctional tertiary or major referral hospital	47	42.7
Availability of snake antivenom in the hospital		
Yes	108	98.2
No	0	
Unsure	0	
Not stated	2	1.8

^1^ ACEM = Australasian College of Emergency Medicine; ^2^ ACRRM = Australian College of Rural and Remote Medicine.

**Table 2 toxins-13-00482-t002:** Types of snake bites and presenting history (*n* = 121).

Item	No.	(%)
Did the patient survive?		
Yes	113	93.4
No	6	5.0
No information	2	1.7
Patient presenting history		
Eyewitness of snakebite	72	59.5
Patient claimed to feel something bite/strike them	45	37.2
No information from patient	0	
Other	4	3.3
Snake identified by reptile handler, herpetologist, zoo or museum snake experts		
Yes	18	14.9
No	102	84.3
Not stated	1	0.83
If snake was identified, type *n* = 18		
Brown Snake	5	27.8
Tiger snake	5	27.8
Taipan	3	16.7
Death adder	4	22.2
Not stated	1	5.6
Symptoms observed at presentation		
Pain at site	57	47.1
Feeling anxious	54	44.6
Headache	51	42.1
Dizziness	31	25.6
Vomiting	33	27.3
Abdominal pain	28	23.1
Blurred vision	18	14.9
Collapse	8	6.6
Chest pain	3	2.5
Others	19	15.7

**Table 3 toxins-13-00482-t003:** Knowledge and attitudes on snake envenoming (*n* = 110).

Item	No.	(%)
Main source of information for managing snake envenoming		
Phone advice	32	29.1
Hospital guidelines	22	20.0
State guidelines	22	20.0
National guidelines	10	9.1
Therapeutic guidelines	8	7.3
Other	14	2.7
Not stated	2	1.8
What would be the first most important factors to influence your decision to use snake antivenom?		
Adverse reaction	0	0
Laboratory findings	37	33.6
Clinical presentation	64	58.2
Antivenom availability	1	0.91
Efficacy of evidence	8	7.3
Cost	0	0

**Table 4 toxins-13-00482-t004:** Snake envenoming case management and antivenom use (*n* = 121).

Item	No.	%
Types of laboratory tests conducted		
20 min WBCT	9	7.4
Fibrinogen	111	91.7
INR	116	95.9
APPT	117	96.7
PT	98	81.0
D-dimer	105	86.8
Serum electrolytes	115	95.0
Renal function	114	94.2
Snake venom detection kit	40	33.1
CK	112	92.6
CBC	102	84.3
Other	7	5.8
Antivenom administered (*n* = 121)		
Yes	67	55.4
No	54	44.6
Type of antivenom dispensed (*n* = 67)		
Polyvalent	13	19.4
Taipan	5	7.5
Tiger snake	31	46.3
Brown snake	31	46.3
Death adder	3	4.5
Sea snake	0	0
Black snake	1	1.5
Number of total vials per suspected case (*n* = 67)		
1 vial	38	56.7
2 vials	23	34.3
3 vials	2	3.0
4 vials	2	3.0
20 vials	1	1.5
Antivenom treatment effective for this case (*n* = 67)		
Strongly disagree	4	6.0
Disagree	5	7.5
No opinion	12	17.9
Agree	25	37.3
Strongly agree	21	31.3
Adverse event after antivenom (*n* = 67)		
Yes	8	11.9
No	59	88.1

**Table 5 toxins-13-00482-t005:** Barriers and learning needs to snake envenoming management (*n* = 110).

Item	Agree, *n* (%)	Uncertain, *n* (%)	Disagree, *n* (%)
What are the barriers to managing snake envenoming?			
Training	63 (57.8)	11 (10.1)	35 (32.1)
Pathology services required for diagnosis	47 (43.9)	7 (6.5)	53 (49.5)
Availability of clinical practice guidelines	31 (29.2)	13 (12.3)	62 (58.5)
Availability of antivenom	25 (23.4)	13 (12.1)	69 (66.3)
I am confident in my ability to…			
diagnose a possible snake bite envenoming	109 (99.1)	1 (0.91)	0 (0)
determine the need of using antivenom	106 (96.4)	3 (2.7)	1 (0.91)
manage adverse reactions related to the use of antivenom	101 (91.8)	8 (7.3)	1 (0.91)
select the appropriate antivenom	87 (79.1)	14 (12.7)	9 (8.2)

**Table 6 toxins-13-00482-t006:** Perceptions on the source of information for managing snake envenoming (*n* = 110).

Item	Agree, *n* (%)	Uncertain, *n* (%)	Disagree, *n* (%)	Unaware of Guideline, *n* (%)
I perceive the following to be useful in guiding the management of snake bite envenoming:				
State guidelines on management of snakebite	97 (88.2)	2 (1.8)	4 (3.6)	7 (6.4)
Hospital protocol	78 (70.9)	11 (10.0)	15 (13.6)	6 (5.5)
Therapeutic guidelines	74 (67.3)	25 (22.7)	6 (5.4)	3 (2.7)
Published literature in peer-reviewed journals	64 (58.2)	29 (26.4)	13 (11.8)	0

**Table 7 toxins-13-00482-t007:** Demographic characteristics of surveyed pharmacists (*n* = 31).

Demographic Characteristics	No.	(%)
Main state of practice		
New South Wales	10	32.0
Victoria	10	32.3
Queensland	5	16.1
South Australia	4	12.9
Northern Territory	2	6.5
Emergency department ACEM classification		
Level 1: within a designated area of a remote or rural hospital	4	12.9
Level 2: part of a secondary hospital	6	19.4
Level 3: part of a major regional, metropolitan, or urban hospital	13	41.9
Level 4: part of a large, multifunctional tertiary or major referral hospital	8	25.8
Availability of restocking system to identify expired/near expired antivenom		
Yes	27	87.1
No	4	12.9

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
