# Peer review of "The Snake Study: Survey of National Attitudes and Knowledge in Envenomation"

_toxins, 2021, doi:10.3390/toxins13070482_

Round 1
Reviewer 1 Report
According to my opinion, the manuscript in the present form does not fit well with the journal’s scope, although it deals with management of snakebites in Australia, specifically, concerns and controversies associated with treatment protocol, since there is no data related to whether current clinical practice adheres to available guidelines. The authors conducted the survey of clinicians who had treated envenoming over the last few years and provided information regarding their experience in envenoming treatment. Maybe I have missed the point, but scientific significance of the investigation is poor. The study only gives insight into the current state in the field regarding knowledge, attitudes and practices of the antivenom use. No comparison with available guidelines (with the exception of that concerning antivenom dose) or recommendations about the future ones was given. Briefly, contrary to expectations, the foundation for further application of the results in real environment, enabling shift towards improvement of the medical expertise in treatment of snakebite victims, has not been provided.
Update:
as I have already mentioned, the manuscript has been written in the form of report, and discussion is the most critical since the interpretation of data is poor. According to my opinion, the readers would probably be interested to know how the current practice in real clinical setting follows the guidelines, does the current ones need an improvement in order to achieve better coping of medical stuff with envenoming and if so, what novelties authors suggest based on collected information that might provide clearer instructions for behaviour in emergency situations.
Hope this might help.
Author Response
The survey was conducted after concerns were expressed following a coronial enquiry into two snakebite deaths in Victoria. In seeking expert opinion the coroner noted a lack of consensus in treatment guidelines amongst three experts, all of whom are well respected and two of whom are well published in the area. The coronial enquiry and the subsequent directive to review snakebite guidelines prompted our study. If experts could not agree, how were front line clinicians managing snake bite and what was guiding their care?
Australia has had a long and esteemed history of snakebite research, led over a thirty year period from 1960 by Stuhan Sutherland, who also pioneered the use of the pressure immobilisation bandage for the treatment of Australian elapids and the funnel web spider (Atrax Robustus) in the late 1970s. His contribution was acknowledged in a publication in Toxicon in 2006. (Tibballs J. Struan Sutherland--Doyen of envenomation in Australia. Toxicon. 2006 Dec 1;48(7):860-71. doi: 10.1016/j.toxicon.2006.07.021. Epub 2006 Jul 15. PMID: 16920170).
Our study demonstrates lack of consensus in care. Now that we have shown that the coroner’s concerns are justified, the next step is to rebuild consensus amongst clinicians who treat snake bite and who write guidelines and direct public policy.
As noted in our discussion, over the last two decades, there has been significant controversy over the number of ampoules of antivenom required for the initial dose in treating snake envenoming in Australia following the publication of the Australian Snakebite Project (ASP) study, an invitro venom/antivenom neutralisation study concluded that one ampoule of tiger snake antivenom appeared to be sufficient to bind all circulating tiger snake venom. (Isbister GK, O'Leary MA, Elliott M, Brown SGA. Tiger snake (Notechis spp) envenoming: Australian Snakebite Project (ASP-13). Med J Aust. 2012;197(3):173-7.)
As clinical toxicologists we believe it is important to assess the impact of the evolving evidence (some of which conflicts with earlier guidance) on the management of snakebite in order to better inform care. We believe that inviting clinicians who have treated a snakebite to participate in the study would be a better way to assess real world experience. The number of clinicians is small, as the number of clinically significant snake bite is small. For example in Victoria, a state of 6.681 million people there were only 128 calls to the Victorian Poison Information Centre in 2020, of whom only 31 were symptomatic at the time of the call.
Our data confirms variation in management. Over one third of respondents stated that they were uncertain or disagreed that publications in peer-reviewed journals were useful in guiding the management of snake bite envenoming. While two thirds of respondents felt that the availability of guidelines was not a barrier, a third were uncertain or agreed they were a barrier to snake bite management. As documented in the paper different number and combination of antivenom was administered in the 67 patients who received antivenom which highlights the concerns and the premise upon which the survey was conducted.
In response to Reviewer 1’s concern that there is no comparator guidelines the survey was conducted to establish how clinicians accessed information currently. Our aim was to determine if there was a need for further work on information sources and we have shown that this is the case. More work needs to be done to align practice across the country. We have not set out to provide the medical expertise (and consensus) but to establish the need to do this.
We have attempted to interpret our data as comprehensively as our sample size would allow. However, we do believe that there is sufficient robustness and consistency of the themes that have emerged to support our discussion and conclusion. If there is a particular aspect of the data interpretation that reviewer 1 would like us to address we would be happy to undertake additional analysis.
The study was not designed to provide clearer instructions on behaviour in emergency situations – it was to ascertain current treatment and sources of knowledge and referral. Further than this was outside the scope of the study.
Finally, the reviewer is concerned that the article does not fit well within the parameters of the Journal. Readers will be well versed with the works of Professors Stuhan Sutherland, Geoffrey Isbister and Julian White, the latter two providing conflicting expert opinion to the coroner in the sentinel case mentioned earlier. The journal has published many article on Australian envenomation and we believe this study would be of interest as well.
On its editorial page Toxicon lists within its scope;
- articles containing the results of original research on problems related to toxins derived from animals, plants and microorganisms
- articles on the translational application of toxins, for example as drugs and insecticides
- review articles on problems related to toxinology.
We believe we have hit the brief by demonstrating that research does not translate well into guidelines when experts differ. Studies such as ours are important in highlighting these issues and raising awareness of problems related to toxinology. If we do not investigate concerns we may not appreciate what issues confront our clinicians and therefore cannot address variation in care.
Reviewer 2 Report
Study is limited by the small number of respondents. Only those who have treated a snakebite in the last 36 months are eligible. Do you have any estimates on the possible number of venomous snake bites in Australia from calls to poison centers or databases on emergency department visits, even if it is only Victoria?
Author Response
Thank you for your review. In response to your comments about snake bite prevalence
Snake bite in Australia leads to approximately 550 annual admissions to public hospitals (2.4 per 100,000) and an average of 2 deaths per year (Welton, R.E., Liew, D., Braitberg, G., 2017. Incidence of fatal snake bite in Australia: A coronial based retrospective study (2000–2016). Toxicon 131, 11–15. DOI:10.1016/j.toxicon.2017.03.008). Since 2007 there have been 1054 calls to the Victorian Poison Information Centre relating to snake bite exposures, an average of 81 patients per year. In the 2020 calendar year there were 128 calls regarding snake bite exposure of which 63 patients were reported to be asymptomatic at initial call, 31 were reported to be symptomatic (all but four reporting minor symptoms), while symptoms were not reported or documented as unknown in thirty four patients. Of this cohort, 72 were male, 54 female and two unknown.
The Victorian Poison information Centre serves a population of 6.681 million (as of September 2020) hence the call rate is 0.001%
Round 2
Reviewer 1 Report
The need for conducting the study in the form of survey has now been explained more argumentatively. Also, the goal, summarized in the middle paragraph of the Conclusions section, has now been clearly stated. As a suggestion, maybe it would be a good idea to place it at the end of the introduction. I accept all the changes in the new version of the manuscript and reasons, listed in the Authors’ responses to reviewer’s comments, explaining why the comparison with available guidelines and recommendations about the future ones were out of the scope of the investigation. In general, the paper is nicely written and easy to follow. I hope it will encourage the regulatory bodies to standardize the protocol for the envenoming treatment.

Author Response
Thank you for review and comments. I have taken your suggestion and taken the middle paragraph of the Conclusions section and have placed it at the end of the introduction.